# Mechanisms Involved in the Relationship between Low Calcium Intake and High Blood Pressure

**DOI:** 10.3390/nu11051112

**Published:** 2019-05-18

**Authors:** Cecilia Villa-Etchegoyen, Mercedes Lombarte, Natalia Matamoros, José M. Belizán, Gabriela Cormick

**Affiliations:** 1Laboratory of Cardiovascular Surveillance of Drugs, Department of Toxicology and Pharmacology, School of Medicine, Universidad de Buenos Aires, Ciudad Autonoma de Buenos Aires, Buenos Aires 1121, Argentina; 2Bone Biology Laboratory, School of Medicine, Rosario National University, Rosario, Santa Fe 3100, Argentina; mercedes_lombarte@yahoo.com.ar; 3Instituto de Desarrollo e Investigaciones Pediátricas “Prof. Dr. Fernando E. Viteri” Hospital de Niños “Sor María Ludovica de La Plata (IDIP), Ministerio de Salud/Comisión de Investigacines Científicas de la Provincia de Buenos Aires, La Plata, Buenos Aires 1900, Argentina; natymatamoros@gmail.com; 4Department of Mother and Child Health Research, Institute for Clinical Effectiveness and Health Policy (IECS-CONICET), Ciudad Autonoma de Buenos Aires, Buenos Aires 1414, Argentina; belizanj@gmail.com (J.M.B.); gabmick@yahoo.co.uk (G.C.); 5Departamento de Salud, Universidad Nacional de La Matanza, Florencio Varela, San Justo 1903, Argentina

**Keywords:** calcium intake, blood pressure, parathyroid function, vitamin D, renin-angiotensin-aldosterone system

## Abstract

There is increasing epidemiologic and animal evidence that a low calcium diet increases blood pressure. The aim of this review is to compile the information on the link between low calcium intake and blood pressure. Calcium intake may regulate blood pressure by modifying intracellular calcium in vascular smooth muscle cells and by varying vascular volume through the renin–angiotensin–aldosterone system. Low calcium intake produces a rise of parathyroid gland activity. The parathyroid hormone increases intracellular calcium in vascular smooth muscles resulting in vasoconstriction. Parathyroidectomized animals did not show an increase in blood pressure when fed a low calcium diet as did sham-operated animals. Low calcium intake also increases the synthesis of calcitriol in a direct manner or mediated by parathyroid hormone (PTH). Calcitriol increases intracellular calcium in vascular smooth muscle cells. Both low calcium intake and PTH may stimulate renin release and consequently angiotensin II and aldosterone synthesis. We are willing with this review to promote discussions and contributions to achieve a better understanding of these mechanisms, and if required, the design of future studies.

## 1. Introduction

The relationship between calcium intake and blood pressure has been widely studied since the 1980s [1,2,3]. Dietary calcium has been shown to have an effect on blood pressure in animal studies. Normotensive rats fed a free-calcium diet significantly increased their systolic blood pressure (SBP) between 15 to 35 mmHg in comparison with rats fed with normal calcium diet [4,5,6]. On the other hand, normotensive and hypertensive rats supplemented with calcium had significant lower values of SBP [7,8,9,10]. Systematic reviews of calcium supplementation randomized controlled trials in hypertensive and normotensive populations have shown a consistent decrease of blood pressure, with a mean difference in systolic blood pressure (SBP) of 2.5 mmHg (95% confidence interval (CI) = 0.6–4.5) in hypertensive subjects and of 1.4 mmHg (95% (CI) = 0.72–2.15) in normotensive subjects [11,12]. In humans, even such a small reduction in blood pressure was estimated to be associated with about 10% lower stroke mortality and about 7% lower mortality from ischemic heart disease [13].

The effect of calcium supplementation on SBP was higher in people aged less than 35 years (−2.11 mmHg) and with doses of calcium equal to or over 1500 mg/day (−2.79 mmHg). The higher impact on BP reduction observed in these cases seem to be diluted in the overall revision due to these studies only representing approximately 20% of the participants [11]. Also, the vast majority of the included trials were performed in high-income countries that usually have a higher basal dietary calcium intake.

Most importantly this effect on blood pressure has been studied during pregnancy for the prevention of preeclampsia [14]. A systematic review of randomized controlled trials of calcium supplementation on the prevention of preeclampsia shows a large effect (13 trials, 15,730 women: RR = 0.45, 95% CI = 0.31–0.65) [15]. This evidence was used to update WHO guidelines for the prevention of preeclampsia, which include the recommendation to supplement pregnant women from areas with low calcium intake with 1.5 to 2 g of calcium per day during the second half of pregnancy [16,17]. Moreover, the follow up of children whose mothers were supplemented during pregnancy show noticeable effects on preventing high blood pressure and dental caries in the progeny [18,19]. Recently, the preconceptional and early pregnancy effect of a low calcium supplementation dose was studied in a multi-country randomized placebo controlled trial showing the beneficial effect of calcium before conception and throughout pregnancy [20]. All these effects of calcium intake have also been replicated in animal models so as to gain insight on the mechanisms that link calcium intake and blood pressure regulation [1,2,4,5,6].

The aim of this literature review is to contribute to finding the mechanisms that could explain the relationship between calcium intake and blood pressure. 

## 2. Calcium and Blood Pressure Regulation

Calcium intake may regulate blood pressure by increasing intracellular calcium in vascular smooth muscle cells leading to vasoconstriction, and by increasing vascular volume through the renin–angiotensin–aldosterone system (RAAS). We found three major mechanisms explaining the relationship between a low calcium intake and the increase in blood pressure: (a) parathyroid function, (b) vitamin D, and (c) the renin–angiotensin–aldosterone system (RAAS). These three mechanisms are described below (Figure 1).

### 2.1. Calcium Intake and Parathyroid Function on Blood Pressure Regulation

Calcium intake has a role in blood pressure regulation and the parathyroid glands play a role in calcium homeostasis, thus the link between calcium intake, parathyroid function, and blood pressure seems to be intuitively valid. However, few studies have evaluated the pathways linking calcium intake, parathyroid function, and blood pressure altogether. The following section accounts for the physiological bases of this relationship, focusing on the effects of the parathyroid hormone (PTH) and the not completely purified and characterized parathyroid hypertensive factor (PHF).

#### 2.1.1. Mechanisms Mediated by Parathyroid Hormone (PTH)

Several studies have reported that calcium intake is inversely associated with blood pressure, both in humans [3,11,12,21,22,23] and in animals [2,5,6]. The inverse relationship between calcium intake and plasma PTH levels has also been widely studied at different ages and physiological stages in both humans and animals, in acute and long term studies [24,25,26,27]. Similarly, the direct relationship between plasma PTH and blood pressure both in healthy subjects and animals is also well documented [25,28,29,30,31,32,33]. The relationship between the reported parathyroid hormone serum levels and blood pressure measurements in normotensive and hypertensive subjects is shown in Table 1. Studies showing a correlation between blood pressure values and quartiles and quintiles of parathyroid hormone levels in human studies are shown in Table 2. Some studies in humans have shown low calcium intake, increased PTH levels, and increased blood pressure in the same subjects, although no description of mechanisms involved in this relationship were mentioned [34,35,36] (Table 3). 

Several studies have shown that PTH levels can independently predict cardiovascular disease and mortality [35,37,38,39]. The prospective and multicenter Osteoporotic Fractures in Men (MrOS) study, including 1490 men older than 65 years and followed up for 7.3 years, shows that concentrations of PTH were associated with an increased risk of cardiovascular mortality (adjusted hazard ratio SD = 1.21, 95% CI = 1.00–1.45) and from all causes (adjusted HR per SD = 1.15, 95% CI = 1.03–1.29) [37]. The Multi-Ethnic Study of Atherosclerosis (MESA) cohort study in which 3002 men and women, aged 59 ± 9.7 years, without cardiovascular antecedents were followed up for 9.0 years found that higher PTH serum concentrations were associated with a greater risk of hypertension, even after adjusting for potential confounders (HR = 1.27, 95% CI = 1.01–1.59) [39]. The population-based, cross-sectional Tromsø study including 3570 men and women aged 25–79 years found a positive relationship between serum PTH and SBP with the highest quartile of serum PTH found to be an independent predictor of coronary heart disease in both sexes (OR = 1.70, 95% CI = 1.08–2.70 for males, OR = 1.73, 95% CI = 1.04–2.88 for females, *p* < 0.05). 

PTH is an 84-amino-acid polypeptide (≈9.5 kilodalton (kDa)) secreted by the chief cells of the parathyroid gland. PTH acts on the cell membrane of its target tissues through a G-protein-coupled receptor, the parathyroid hormone receptor type 1 (PTHr-1). Expression of PTH receptors has been reported in many tissues, including vascular smooth muscle and endothelial cells [40]. The PTHr-1 couples to several signaling pathways, namely: the Gαs/adenylate cyclase (AC)/cAMP/protein kinase A (PKA), the Gαq/phospholipase C (PLC)β/inositol trisphosphate (IP3)/intracellular Ca/protein kinase C (PKC), the Gα12/13 phospholipase D/RhoA pathway, and the mitogen-activated protein kinase (extracellular signal regulated kinase [ERK1/2]) signaling cascade [41,42].

Several mechanisms have been proposed to explain the effect of PTH on blood pressure: (a) an increase in cytosolic free calcium concentration ([Ca^2+^]_i_) through the PTH receptor (PTHr-1) in vascular smooth muscle, (b) increase calcitriol concentration, and (c) a cross-talk with the renin–angiotensin–aldosterone system (RAAS). The last two will be described in its corresponding sections.

High [Ca^2+^]_i_ increases vascular reactivity, and therefore peripheral vascular resistance and responsiveness to the sympathetic and the RAAS, which all elevate blood pressure. Calcium channel blockers, such as nifedipine and verapamil, are valuable antihypertensive drugs as they inhibit Ca^2+^ entry to the cell and reduce [Ca^2+^]_i_. In the same way, calcium supplementation in subjects with low calcium intake has been described to decrease [Ca^2+^]_i_ [43,44], hence diminishing blood pressure. It has been shown that PTH increases calcium entry into a variety of mammalian tissues and cell lines, such as cardiomyocytes [45], enterocytes [46], kidney [47], liver [48], peripheral nerves [49], osteosarcoma cells [50], and osteoblastlike cells [51]. Significantly higher [Ca^2+^]_i_ was also found in human platelets and lymphocytes of hypertensive patients [29,52]. The activation of PTHr-1/Gq/PLC/IP3, PTHr-1/Gα12-13/phospholipase D/RhoA cascades, and of calcium channels are the signaling pathways by which PTH increases [Ca^2+^]_i_ and blood pressure [47].

A controversial effect is the vasodilator effect of acute PTH infusion, both in vivo and in vitro. In vascular smooth muscle cells, PTHr-1 couples primarily to Gαs, which increases cAMP and decreases [Ca^2+^]_i_ [53]. Nonetheless, the sustained activation of this cascade shows desensitization to PTH in a time- and concentration-dependent fashion [54,55,56]. The chronic infusion of PTH has been associated with arterial hypertension [57]. Long-standing high levels of PTH, such as in hyperparathyroidism, are frequently related to hypertension, whereas parathyroidectomy is associated with a decrease in [Ca^2+^]_i_ and blood pressure [58]. A rise of [Ca^2+^]_i_ through PTHr-1/Gαs/AC/cAMP via opening calcium channels in a cell line derived from fetal rat aorta was also described [59]. Therefore, the desensitization of the cAMP pathway to PTH, as well as the stimulation of other blood pressure mediators considered below, like the RAAS and calcitriol, may explain the long-term pressor effects of PTH.

#### 2.1.2. Parathyroid Hypertensive Factor (PHF)

In the early 1990s, Lewanczuk et al. described that the infusion of plasma from hypertensive rats and from hypertensive subjects on normotensive rats increased the mean arterial pressure of those rats [60,61]. They attributed this effect to the presence of a novel hypertensive factor in the plasma of the hypertensive donors. The same group also reported the parathyroid origin of this factor by transplanting parathyroid glands from hypertensive rats into parathyroidectomized normotensive rats. An increase in blood pressure was shown in the rats after transplantation [60,62,63]. Due to this, the non-isolated substance was called parathyroid hypertensive factor (PHF) [60]. 

In spontaneously hypertensive rat strains, low calcium intake increases blood pressure that the authors explained was due to an increase of PFH [64]. These authors also proposed that PHF regulates blood pressure by modifying the concentration of [Ca^2+^]_i_ in vascular smooth muscle [60,65,66]. In isolated vascular smooth muscle cells from rat tail arteries, Shan et al. found that the infusion of semi-purified plasma from hypertensive rats enhanced the opening of the L-type calcium channel, an effect antagonized by the dihydropyridine calcium channel blocker nifedipine [67]. 

Although the effect of plasma from hypertensive rats has been well-documented, the purification and characterization of PHF is uncomplete. Benishin et al. showed the correlation of a UV spectrum of dialyzed plasma from hypertensive rats with a small (≈3 kDa), trypsin-inactivated and boiled-resistant peptide [68]. Years later, the same group proposed that the PHF structure has both peptide and lysophospholid motifs, critical for its biological activity [69]. Schlüter et al. also isolated a peptide-like vasopressor of low molecular weight (0.6–2.5 kDa) from parathyroid tissue of patients with tertiary hyperparathyroidism. The eluent was shown to be polar, hydrophilic, and protease-sensitive, but not heat-resistant. Schlüter’s group also suggested that further purifications were needed as several substances were shown on the mass spectrometry [70]. 

Although the existence of PHF may explain the blood pressure changes induced by alterations in a calcium diet, there is still much to know about this mediator. First, the published results have been poorly replicated outside the group that described PHF [71]. On the other hand, after 30 years of being described, the definitive chemical structure of this mediator is still unknown and there have not been reports of new attempts to purify it. Nonetheless, the function of the parathyroid gland seems to play a key role in blood pressure regulation, as was reported both in humans and in normotensive and hypertensive animals.

In summary, the bulk of information of this section orients towards an increase of the parathyroid gland activity by low calcium intake leading to an increase of [Ca^2+^]_i_ in vascular smooth muscle cells and consequently a rise in blood pressure (Figure 1). In addition to the direct effect of PTH (or PHF) in smooth muscle, we will later discuss the effects of PTH mediated by calcitriol and RAAS. A demonstration of the effect of parathyroid activity in the regulation of blood pressure related to calcium intake is shown in a study of parathyroidectomized rats in comparison with sham operation rats. After 10 weeks on a calcium-free diet, the sham operation rats showed an increase in SBP of 3.44 (SE = 1.95) mmHg, while the parathyroidectomized rats showed a decrease in SBP of −9.67 (SE = 2.05) mmHg. A highly statistical significant difference of 13.11 mmHg was found between the two groups [2].

### 2.2. Calcium Intake and Vitamin D in Blood Pressure Regulation

Most systematic reviews of randomized controlled trials do not show an effect of vitamin D supplementation on blood pressure. A systematic review of 46 trials (4541 participants) found no effect of vitamin D supplementation on SBP (effect size = 0.0 [95% CI = −0.8 to 0.8] mmHg; *p* = 0.97; I^2^ = 21%) or diastolic blood pressure (DBP) (effect size = −0.1 [95% CI = −0.6 to 0.5] mmHg; *p* = 0.84; I^2^ = 20%) nor in subgroup analysis via basal vitamin D or blood pressure levels [72]. A systematic review in hypertensive subjects did not find evidence of vitamin D supplementation on blood pressure [73]. However, one systematic review found that in vitamin D deficient populations, supplementation has a small effect on peripheral DBP but not in central DBP nor in peripheral nor central SBP, though this evidence is weak as it mainly comes from one small trial on vitamin D deficient and hypertensive subjects [74,75]. However, in these systematic reviews some population groups, such as those with vitamin D deficiency, might be underrepresented. One randomized, double-blind, placebo-controlled clinical trial of oral calcitriol in African-Americans with low baseline calcitriol level reported a significant decrease in SBP but not DBP after a three-month follow up [76].

Vitamin D is a steroid hormone that can be synthesized in the skin under the influence of sunlight (cholecalciferol or vitamin D3) or be obtained from foods such as fish and vegetables (ergocalciferol or vitamin D2). Later, a two-step hydroxylation is required to procure its biological active form (1,25-OHVitD, calcitriol). Low calcium intake increases calcitriol concentration, both in human and in animals [77,78,79]. PTH stimulates the renal α-1 hydroxylase enzyme to produce the final activation. However, PTH is probably not the sole regulator of calcitriol synthesis as an increase in the efficiency of calcium absorption has been described with low calcium intake diets even after parathyroidectomy [79]. The renal α-1 hydroxylase activity seems to be directly modulated by calcium, as the addition of calcium ions in vitro produced an inhibitory effect of the enzyme activity [80]. In addition, α-1 hydroxylase activity was found in tissues other than kidney such as endothelial cells. Vitamin D receptor (VDR) expression was described in endothelial cells. These findings could orient towards an auto/paracrine action of calcitriol in endothelial cells [81].

The effects of calcitriol are mediated by both genomic and nongenomic mechanisms. Genomic responses are mediated by intracellular vitamin D receptor (VDR) functioning as transcription factors to modulate gene expression, whilst the short-term effects seem to be mediated by putative receptors [82]. 

Among the rapid nongenomic effects, it has been shown that calcitriol increases [Ca^2+^]_i_ by increasing calcium uptake in many types of cells, including rat vascular smooth muscle cells and aorta-derived cell lines [83,84,85]. The intraperitoneal administration of calcitriol on spontaneously hypertensive and normotensive rats enhanced the contractile response of isolated mesenteric resistance arteries, showing that the effect is not related to the hypertensive condition [86]. Furthermore, Shan et al. reported a significant increase in the calcium channel current of rat artery-derived smooth muscle cells after calcitriol infusion [87]. In skeletal muscle, calcitriol effects have been extensively studied showing that calcitriol: (a) stimulates L-type voltage-operated calcium channels by cAMP pathway, (b) activates AC/cAMP/PKA and PLC/IP3/PKC signaling cascades, and (c) boosts the calcium messenger system [88], all of which are responsible for the increase of [Ca^2+^]_i_ (Figure 1).

Calcitriol regulates RAAS via genomic mechanisms [89]. An inverse association between serum calcitriol levels and plasma renin activity, in both human and animals, has been described [90]. Studies in VDR knockout mice show that calcitriol negatively regulates the RAAS and blood pressure by a genomic calcium- and PTH-independent mechanism [91,92,93].

The mechanism by which calcitriol regulates blood pressure seems not to be the same with low calcium intake as it is with vitamin D deficiency. Whereas restrictive calcium diets may dominate the nongenomic short-term effects by increasing [Ca^2+^]_i_, vitamin D deficiency may preponderate the VDR mediated actions on the RAAS.

In summary, the mechanism by which calcitriol regulates blood pressure in low calcium diets may be mediated by the nongenomic short-term effects increasing [Ca^2+^]_i_ and consequently rising blood pressure.

### 2.3. Calcium Intake and Renin–Angiotensin–Aldosterone System

The renin–angiotensin–aldosterone system (RAAS) plays a key role in the physiologic regulation of blood pressure. As shown below, several interactions between calcium intake, calcium homeostatic hormones and the RAAS have been described. 

#### 2.3.1. Renin

Renin enzymatic activity consists in the hydrolysis of angiotensinogen to angiotensin I. The acute incubation or infusion of high concentrations of calcium has an inhibitory effect on renin secretion, both in isolated cells and animals [94,95,96,97]. Dietary calcium supplementation has shown to decrease but not to abolish the renin release [98]. It has been reported that acute activation of the calcium sensing receptor in juxtaglomerular cells by high extracellular calcium concentration decreases renin release and plasma renin activity, both in vivo and in vitro. Renin secretion is dependent on cAMP formation. Increased extracellular calcium concentration decreases renin release by diminishing AC and enhancing phosphodiesterase activities [95,99]. cAMP concentration could also be raised by PTH as PTHr-1 along the nephron, including the juxtaglomerular apparatus [100]. The stimulation of PTHr-1 increases cAMP and enhances renin release [101] (Figure 1). 

#### 2.3.2. Angiotensin II

Angiotensin II is the primary active product of the RAAS and a potent vasoconstrictor whose actions are mediated by the type I angiotensin II receptor (AT-1R). Low calcium diets have been shown to increase angiotensin II [6,102]. In animals fed low calcium diets, the binding of angiotensin II withAT-1R decreases in smooth muscle cells and increases in the adrenal cortex. These findings suggest that with low calcium intake, angiotensin II may raise blood pressure via increasing aldosterone synthesis or secretion [5]. 

It has also been described that the infusion of angiotensin II produced a significant dose-dependent increase in PTH serum levels [101]. Moreover, AT-1R has been isolated in the parathyroid glands, and although the intracellular pathway is still unknown, AT-1R inhibition lowered PTH levels [103]. These observations may explain the blood pressure elevation by angiotensin II in a low calcium diet (Figure 1).

#### 2.3.3. Aldosterone

Aldosterone, a steroid hormone, is synthesized by the zona glomerulosa of the adrenal cortex (CZG) in the adrenal gland. Its primary effect is the regulation of blood pressure through the reabsorption of sodium and excretion of potassium in the distal tubules and the collecting ducts of the nephron [104], increasing apical membrane permeability for sodium, thus causing sodium and water reabsorption. This rise of extracellular fluid volume increased cardiac output, and hence blood pressure (Figure 2). 

Aldosterone synthesis is stimulated by angiotensin II and high extracellular potassium levels via the calcium messenger system that boosts the steroidogenic cascade within the mitochondria. The acute steroidogenic regulatory protein (StAR) is a key molecule that transports cholesterol through the inner mitochondrial membrane, a fundamental step for the synthesis of steroid hormones, such as aldosterone. StAR is activated by increases in [Ca^2+^]_i_. Angiotensin II, via AT-1R activating the PLC/IP3 cascades, and high extracellular potassium levels via depolarizing the plasma membrane of the cell activating calcium voltage-dependent channels, increase [Ca^2+^]_i_ and thus aldosterone synthesis [105].

A bidirectional stimulating relationship between PTH and aldosterone is reported. Patients with primary aldosteronism have high levels of PTH that return to physiological concentration after the adenoma removal or pharmacological treatment with aldosterone antagonists [106]. Similarly, hypertension associated with secondary aldosteronism due to hyperparathyroidism resolves after parathyroidectomy [107,108]. Studies in humans have also shown a rise in blood pressure and aldosterone after two weeks of an infusion of PTH [57], and this effect is antagonized by PTH-receptor blocking agents [109]. PTHr-1 has been found in the CZG of animals and humans [110], and mineralocorticoid receptors, for aldosterone, have been identified in the parathyroid gland [111]. 

It has been shown that PTHr-1 stimulates basal steroid secretion (aldosterone, cortisol) from adrenal cells through both AC/PKA- and PLC/PKC-dependent signaling mechanisms [109]. Adrenal cells incubated with PTH increase cAMP and IP3 production, with this effect being partially suppressed by inhibitors of both cascade. The activation of PKA, by increasing the concentration of cAMP, activates StAR. PTH can also indirectly favor steroidogenesis by increasing [Ca^2+^]_i_ [109]. 

In summary, it has been shown that both low calcium intake and PTH stimulate renin release, and consequently, angiotensin II and aldosterone synthesis. 

## 3. Conclusions

In this manuscript, we reviewed the literature exploring the mechanisms involved in the relationship between calcium intake and blood pressure. It has been shown that, particularly in individuals with low calcium intake, an increase in calcium intake reduces blood pressure. We consider that in view of hypertension being a major factor involved in the global burden of disease, the study of interventions that could prevent the development of hypertension should be prioritized [112]. The link between calcium intake and blood pressure involves a connection between calciotropic hormones and blood pressure regulators. As was hypothesized many years ago, parathyroid activity increases the cytosolic concentration of calcium and increases vascular reactivity and blood pressure [113]. The effect of calcium intake on blood pressure is not shown in parathyroidectomized animal studies. Low calcium intake also increases the synthesis of calcitriol in a direct manner or is mediated by PTH. Calcitriol increases intracellular calcium in vascular smooth muscle cells. Low calcium intake stimulates renin release, and consequently, angiotensin II synthesis. PTH stimulates renin release, angiotensin II and aldosterone synthesis (Figure 3). We are willing with this review to promote discussions and contributions to achieve a better understanding of these mechanisms, and if required, the design of future studies.

## Figures and Tables

**Figure 1 nutrients-11-01112-f001:**
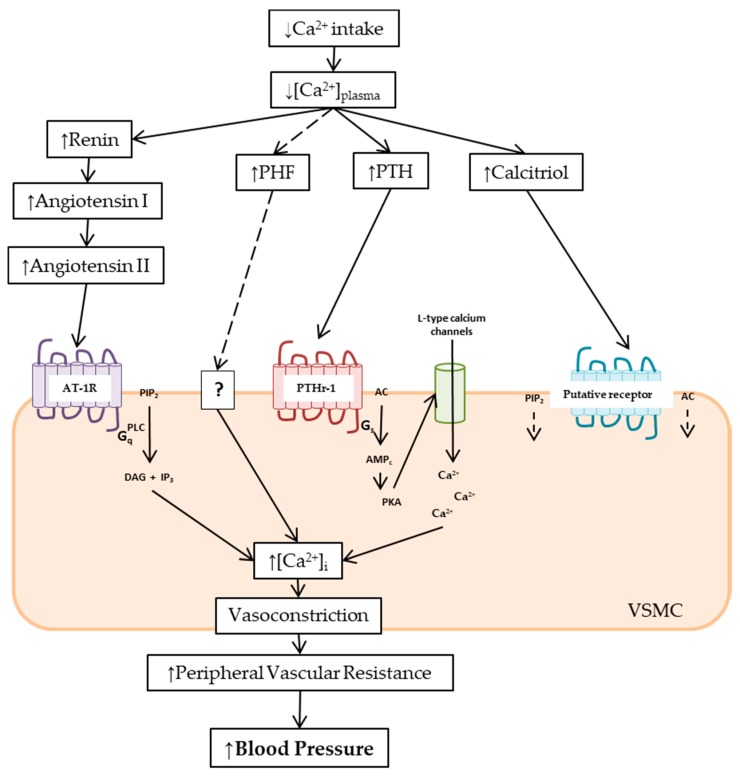
Scheme of the mechanisms involved in the rise of blood pressure in low calcium intake by an increase in peripheral vascular resistance. Low calcium intake decreases plasmatic calcium concentration ([Ca^2+^]_plasma_), stimulating the release of parathyroid hormone (PTH) and parathyroid hypertensive factor (PHF), the synthesis of calcitriol, and the activation of the renin–angiotensin–aldosterone system (RAAS). In vascular smooth muscle cells (VSMC), angiotensin II via the angiotensin II type I receptor (AT1R)/Gq/phospholipase C (PLC)/inositol trisphosphate (IP3) pathway, PTH via PTHr-1/Gs/3′,5′-cyclic adenosine monophosphate (cAMP)/protein kinase A (PKA), and calcitriol via adenylate cyclase (AC)/cAMP/PKA and PLC/IP3 signaling pathways increased the intracellular calcium concentration ([Ca^2+^]_i_). The rise of [Ca^2+^]_i_ leads to vasoconstriction, and hence increases in peripheral vascular resistance and blood pressure. The PHF mechanism of action remains unknown. See text for further details.

**Figure 2 nutrients-11-01112-f002:**
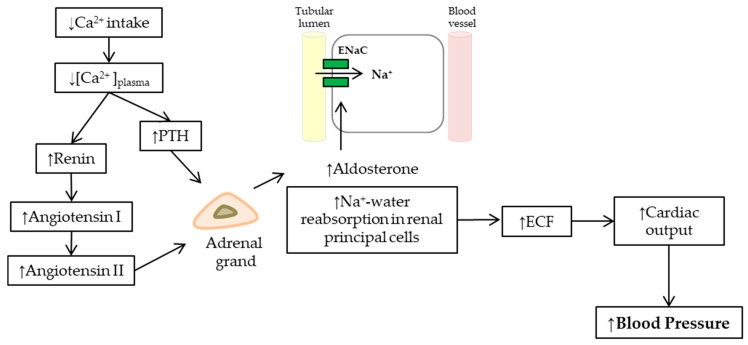
Scheme of the mechanisms involved in the rise of blood pressure in low calcium intake via an increase in cardiac output. Low calcium intake decreased the plasmatic calcium concentration ([Ca^2+^]_plasma_), stimulating PTH and the renin–angiotensin–aldosterone system (RAAS). Both angiotensin II and PTH were increased aldosterone secretion due to the adrenal gland. Aldosterone upregulates epithelial sodium channels (ENaC) in the principal cells of the collecting duct in the kidney, increasing apical membrane permeability for Na^+^, thus Na^+^ and water reabsorption. The rise of extracellular fluid volume (ECF) increased cardiac output and hence blood pressure.

**Figure 3 nutrients-11-01112-f003:**
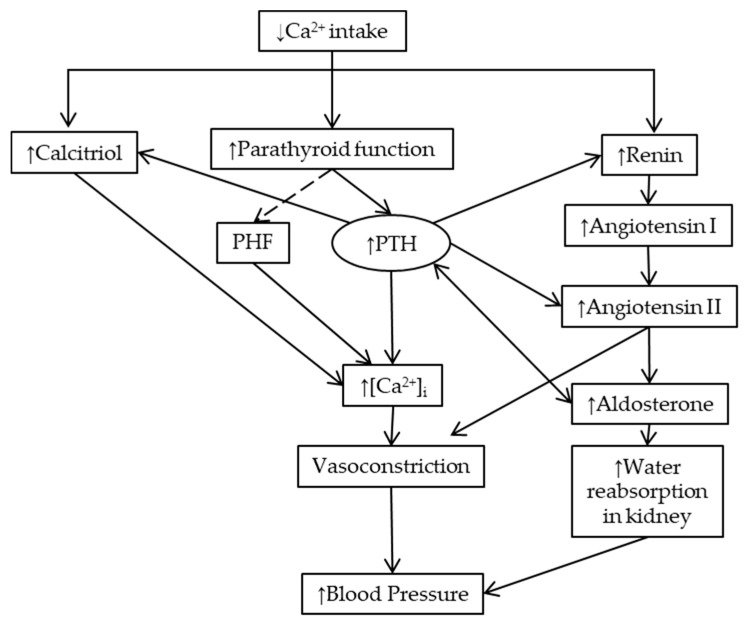
Scheme of the mechanisms involved in the rise of blood pressure due to low calcium intake, namely the link between calciotropic hormones and blood pressure regulators. Low calcium intake: (a) increases calcitriol serum levels, (b) stimulates parathyroid function, and (c) increases renin secretion. Calcitriol may increase cytosolic free calcium concentration ([Ca^2+^]_i_) via non-genomic short-term mechanisms. The parathyroid gland secretes parathyroid hormone (PTH) and possibly (dash arrow) the parathyroid hypertensive factor (PHF). Both mediators increase [Ca^2+^]_i_, leading to the contraction of the vascular smooth muscle cells (vasoconstriction). Renin release is stimulated both by low extracellular calcium and PTH, activating the renin–angiotensin–aldosterone system (RAAS). In addition, PTH increases angiotensin II and aldosterone synthesis, which also leads to vasoconstriction and increases renal water reabsorption, increasing blood pressure. Aldosterone also increases PTH serum levels (double-headed arrow). See text for further details.

**Table 1 nutrients-11-01112-t001:** Parathyroid hormone (PTH) serum levels and blood pressure (BP) values in normotensive and hypertensive subjects.

Reference (First Author)	Method	Country and Participants	PTH (pmol/L)	BP (SBP − DBP mmHg)
Young 1990 [28]	Cross-sectional	USA, 115 subjects, ≈45 years	NT = 4.5 ± 2.2	NT = 120(±11) − 80(±8)
			HT = 5.0 ± 2.4	HT = 138(±8) − 95(±5)
Brickman 1990 [29]	Cross-sectional	USA, 38 men, ≈56 years	NT = 20.8 ± 1.1	NT = 123(±2.8) − 78(±1.3)
			HT = 28.4 + 3.5	HT = 150(±3.9) − 97(±0.9)
Morfis 1997 [30]	Cross-sectional	Australia, 123 subjects, 63–88 years	NT = 2.7 ± 1.1	NT = 125(±12) − 71(±7)
			HT = 2.9 ± 1.3	HT =135(±14) − 73(±10)
Park 2015 [32]	Cross-sectional	Korea, 1664 postmenopausal women, >50 years	NT = 63.7 ± 23.4	NT = 117.5(±12.4) − 73.3(±8.1)
			HT = 68.3 ± 23.6	HT = 149.4(±11.4) − 86.0(±10.1)

NT = normotensive people; HT = hypertensive people, SBP = systolic blood pressure; DBP = diastolic blood pressure. Values are expressed as mean (±standard deviation).

**Table 2 nutrients-11-01112-t002:** Blood pressure (BP) values by quartiles or quintiles of parathyroid hormone (PTH) levels in human studies.

Reference (First Author)	Method	Country and Participants	PTH (pmol/L)	BP (SBP − DBP mmHg)
Snijder 2007 [31]	Cross-sectional	The Netherlands, 1205 subjects, participants, 55–85 years	Q1: <2.45	150.1(±26.1) − 82.5(±13.0)
			Q2: 2.45–3.13	151.7(±24.8) − 82.6(±13.4)
			Q3: 3.14–4.25	154.7(±24.6) – 84.3(±13.6)
			Q4: >4.25	156.2(±27.6) − 83.9(±13.0)
Chan 2011 [25]	Cross-sectional	China, 939 men, >65 years	Q1: <3.1	135.8(±1.7) − 76.5(±0.8)
			Q2: 3.2–4.1	139.9(±1.6) − 76.5(±0.8)
			Q3: 4.2–5.5	141.4(±1.7) − 76.5(±0.8)
			Q4: >5.5	143.6(±1.8) − 79.9(±0.8)
Yao 2016 [33]	Cohort study	USA, 7504 subjects, 45–64 years	Q1: 3.2–28.8	112(±13) − 68.0(±8.2)
			Q2: 28.9–34.9	113(±12) − 68.4(±8.3)
			Q3: 35.0–41.5	114(±12) − 69.4(±8.5)
			Q4: 41.6–50.1	115(±12) − 69.9(±8.1)
			Q5: 50.2–162.6	115(±12) − 70.5(±8.2)

SBP = systolic blood pressure; DBP = diastolic blood pressure. Values are expressed as mean (±standard deviation) or mean [±standard error of the mean].

**Table 3 nutrients-11-01112-t003:** Parathyroid hormone (PTH) and blood pressure (BP) and dietary calcium intake in human studies.

Reference (First Author)	Method	Country and Participants	Ca Intake (mg/day)		PTH (pmol/L)	BP (SBP − DBP mmHg)
Takagi 1991 [34]	Clinical trial	Japan, 9 HT, 65–86 years, CaSup (1 g) vs. diet Ca 500 mg, 8 weeks	1 g/day (CaSup)		27	−13.6 mmHg to −5.0 mmHg
			500 mg/day (diet)		33	−1.5 mmHg to +1.0 mmHg
Jorde 2000 [36]	Cohort study	Norway, 1113 subjects, 30–79 years	592.1(±459.6)		4.5(±1.2)	143.4(±19.9) − 84.3(±10.4)
			400.3(±227.3)		9.1(±2.4)	153.9(±27.1) − 89.7(±14.1)
Kamycheva 2004 [35]	Cross-sectional	Norway, 3570 subjects, >24 years	♂	499(±259)	Q1 <1.9	136.9(±17.7)
				476(±257)	Q2 1.9–2.6	140.1(±19.6)
				443(±233)	Q3 2.61–3.5	142.0(±20.3)
				430(±243)	Q4 >3.5	145.2(±20.3)
			♀	478(±277)	Q1 <1.8	133.2(±18.5)
				428(±227)	Q2 1.8–2.4	135.5(±21.5)
				431(±226)	Q3 2.41–3.3	141.9(±22.4)
				408(±217)	Q4 >3.3	146.5(±23.2)

NT = normotensive people; HT = hypertensive people, CaSup= calcium supplementation; SBP = systolic blood pressure; DBP = diastolic blood pressure. Values are expressed as mean (±standard deviation).

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
