# Peer review of "Mechanisms Involved in the Relationship between Low Calcium Intake and High Blood Pressure"

_nutrients, 2019, doi:10.3390/nu11051112_

Round 1

Reviewer 1 Report

The manuscript covers a long-studied area.  That is, dietary calcium intake, blood pressure and the mechanisms contributing to the rise in BP with depressed dietary calcium intake.  There are several concerns though that this manuscript raises:

Major

1) the BP lowering effect of calcium supplementation is rather modest and borders on undetectable in a clinical environment.  Thus, how important is the calcium-BP effect given the modest average impact?  Are outsized effects buried in the average BP response in certain subgroups?

2) the rationale for reporting individual studies as well as the results of an overall meta-analysis on the same subject, the BP response to dietary calcium, is unclear as some of the small individual studies have a much larger calcium impact on BP than the meta-analysis

3) line 183, the vitamin D studies have significant problems.  Many of these studies did not require vitamin D deficiency by any definition and simply supplemented with vitamin D.  The optimum dose of vitamin D is unknown and has been inadequately studied.  There are studies in blacks showing a significant drop in BP with escalating doses of vitamin D administration.

4) table 1, as laid out currently, is hard to understand

Minor

1) 1 decimal point is enough when displaying BP effects of dietary calcium supplementation

2) line 119, "verapamile" should be "verapamil"

3) line 161, "motives" should be "motifs"

4) line 255,"rise" should be "raise"

5) line 243, "Increase" should be "increased"

Author Response

Dear Editor,

We are very grateful for your prompt and excellent contributions done by the reviewers.  We agree with all their comments and consequently we made the changes suggested by them.  These changes significantly improve the quality of the manuscript.

Please find attached a reply to the comments of reviewer 1. 

We look forward for your evaluation of our reply.

With best wishes, 

The Authors

Reviewer 2 Report

Given the role of PTH in Ca2+ homeostasis, the interrelationship between dietary Ca2+ intake, PTH action and blood pressure regulation is an interesting topic. The current review summarized studies relating to how low Ca2+ intake may elevate blood pressure via PTH and RAAS mechanisms. The presentation of the review is excellent and provides a detailed insight into the direct and indirect mechanism involved. I have the following comments:

Major comments:

1.      The readers would greatly benefit from schematic illustrations of various signal transduction mechanisms leading to increases in blood pressure by water retention/ECF volume increase/Cardiac Output increase and vasoconstriction/peripheral vascular resistance secondary to low Ca2+ intake. The authors may include a separate schematic on PTH mechanism and a schematic on RAAS involvement or can make a combined schematic to explain their relationship.

Minor comments:

1.      The title is not attractive and somewhat ambiguous.

2.      The manuscript should be checked for the use of non-standard expression. For example, in lines 130 and 136, the use of the word ‘desensibilization’ is confusing. The authors perhaps meant desensitization and it should be corrected to standard expression.

Author Response

Dear Editor,

We are very grateful for your prompt and excellent contributions done by the reviewers.  We agree with all their comments and consequently we made the changes suggested by them.  These changes significantly improve the quality of the manuscript.

Please find attached a reply to the comments of reviewer 2. 

We look forward for your evaluation of our reply.

With best wishes, 

The Authors

This manuscript is a resubmission of an earlier submission. The following is a list of the peer review reports and author responses from that submission.